# Scalable Crop Yield Prediction with Sentinel-2 Time Series and Temporal Convolutional Network

**Maria Yli-Heikkilä [1,2,*]**, **Samantha Wittke [3,4]**, **Markku Luotamo [5]**, **Eetu Puttonen [3]**, **Mika Sulkava [1]**, **Petri Pellikka [2]**, **Janne Heiskanen [2]** and **Arto Klami [5]**

1   Natural Resources Institute Finland, Latokartanonkaari 9, FI-00790 Helsinki, Finland
2   Department of Geosciences and Geography, University of Helsinki, FI-00014 Helsinki, Finland
3   Finnish Geospatial Research Institute, National Land Survey of Finland, Vuorimiehentie 5,
    FI-02150 Espoo, Finland
4   Department of Built Environment, Aalto University, FI-00076 Espoo, Finland
5   Department of Computer Science, University of Helsinki, FI-00014 Helsinki, Finland
*   Correspondence: maria.yli-heikkila@luke.fi

**Abstract:** One of the precepts of food security is the proper functioning of the global food markets. This calls for open and timely intelligence on crop production on an agroclimatically meaningful territorial scale. We propose an operationally suitable method for large-scale in-season crop yield estimations from a satellite image time series (SITS) for statistical production. As an object-based method, it is spatially scalable from parcel to regional scale, making it useful for prediction tasks in which the reference data are available only at a coarser level, such as counties. We show that deep learning-based temporal convolutional network (TCN) outperforms the classical machine learning method random forests and produces more accurate results overall than published national crop forecasts. Our novel contribution is to show that mean-aggregated regional predictions with histogram-based features calculated from farm-level observations perform better than other tested approaches. In addition, TCN is robust to the presence of cloudy pixels, suggesting TCN can learn cloud masking from the data. The temporal compositing of information do not improve prediction performance. This indicates that with end-to-end learning less preprocessing in SITS tasks seems viable.

**Keywords:** crop production statistics; yield forecasts; object-based; remote sensing; machine learning; agriculture; time series

## 1. Introduction

Food security is one of the longstanding continuing development priorities of the United Nations and has been reaffirmed in the 2030 Agenda for Sustainable Development [1]. However, since the onset of the Agenda in 2015, the number of people in the world affected by hunger has increased, reflecting persistent regional socio-economic inequalities [2]. Multiple factors interact within the food systems to the detriment of food security and nutrition, the major global drivers being conflict, climate variability and extremes, and economic slowdowns and downturns [2,3]. It is also estimated that the minimum calorie requirement to eliminate projected food undernourishment by 2030 will unlikely be attainable due to the competition for crops harvested for various other uses, such as animal feed and crop-based biofuels [4].

One of the precepts to food security is to ensure the proper functioning of food commodity markets and agricultural derivatives. Market disruptions or shocks in crop production such as extreme weather events can occasionally disturb the equilibrium of the price determination of agricultural commodities. This induces food price spikes and volatility that are often triggering social unrest and food crises. When food markets

are provided with timely and accurate information openly conveyed between actors, the societies are better prepared for such disruptions in food supplies [2].

Today, many national, regional, and even global agricultural monitoring systems are operating at a range of scales to collect, analyze, disseminate, and communicate comprehensive agricultural intelligence to decision makers (see, e.g., [5]). Since the 1972 launch of the first Landsat satellite, earth observation (EO) has become an essential component of such agricultural monitoring systems, providing timely, objective, and global coverage information regarding crop acreage and yield. Free and open access to US Landsat [6] and European Sentinel imagery [7], with high temporal and spatial resolution, and deployment of high-performance computing have further accelerated the utilization of EO data. For a review of a multitude of crop yield forecasting techniques, the interested reader is invited to take a look at, e.g., [8,9].

Despite the promise of EO as an applicable source of information for operational use for fine resolution crop yield forecasts, agricultural monitoring systems and national statistical offices publish pre-harvest forecasts on state/country-levels. Nevertheless, accurate sub-national level information about crop production is a prerequisite for local-scale research or policy evaluation due to the typically high spatial variability in agricultural production. In addition, agricultural producers, especially livestock husbandry, are dependent on local primary production chains (seeds and fodder), because local supply reduces logistics costs. Accurate, and publicly available, local level near-real time information about expected crop production therefore helps the agricultural practitioners, markets, and decision makers to react and adjust in the event of disruptions.

For fine resolution (temporal and spatial) crop production information, we need a prediction model that can produce in-season crop yield estimates from rather long time series of observations (e.g., 121 days in this study). In optical remote sensing, the time series are typically irregular and sparse due to occlusion by clouds and overlap between swaths from adjacent orbits at higher latitudes. The observable unit itself in crop growth monitoring is a field parcel that is typically irregular in shape and size. In this setting, there are plenty of avenues for constructing a representation of an observable unit for a prediction model.

We assume that a field parcel or set of parcels are near-homogeneous areas, managed with similar agricultural practices, sown approximately in the same short period, and growing in similar agroclimatic conditions and can therefore be considered as objects instead of independent pixel units. In object-oriented remote sensing, objects are typically represented by the mean [10–13] or median [14] of the reflectance values within the bounds of an object. However, a point estimate of distribution, such as mean or median, is inclined to loose important discriminating information about the samples. Histograms provide a straightforward means for utilizing more information from the reflectance distribution [15–17], and histograms with more than one dimension can be used if the prediction method can handle a larger feature space.

Random forests (RF) [18] is a widely applied machine learning method in remote sensing [19] and specifically in agricultural monitoring tasks (e.g., [20–24]). In supervised regression tasks, the method aggregates predictions from several randomized decision-trees by averaging. RF is generally recognized for its high performance, easy parametrization and robustness, and its ability to work in high-dimensional feature spaces while having relatively low sample sizes compared with neural networks that usually perform better with larger sample sizes. However, RF does not per se capture the temporal dimension. Each time step is merely added to the input data as a new independent static feature. This also implies that for each time step of interest (for each feature set), a separate model needs to be trained, making it impractical for automatized near-real time monitoring.

The resurgence of neural networks and resulting advances in computer vision and deep learning inspired by, e.g., LeCun et al. [25] have given rise to novel applications in various domains, including remote sensing [26]. The development of deep neural network models (DNN) has increasingly enabled end-to-end learning of subtle latent features, which

previously required separate extraction. DNNs can reduce manual feature extraction in a variety of fields, ranging from computer vision to chemometrics or assisting clinical decisions [27–29]. In agricultural monitoring, DNNs have been applied for remotely sensed time series, e.g., by [12,15,30–35].

A recent novelty of the temporal convolutional network (TCN) is a sequential implementation of convolutional neural networks (CNN) that uses dilated convolutions. The term was first introduced by [36] in their study on action segmentation and detection. Ref. [37] showed that TCNs could outperform canonical recurrent neural networks (RNN) for sequence modeling tasks on diverse benchmark datasets. Compared to RNNs, TCNs can have a very long effective history making them a favorable option for crop monitoring. TCNs also have the computational advantage of processing convolutions in parallel instead of sequentially as in RNN. Ref. [37] In agricultural monitoring, [38] applied TCN to crop classification. Ref. [39] used TCN in combination with RNN for greenhouse crop yield forecasting. Ref. [40] applied an adaptation of TCN by introducing a channel (band) attention mechanism in crop classification.

Occlusion by clouds in optical satellite images inevitably requires a handling strategy for most remote sensing tasks and data. In most applications, cloud detection is involved as an isolated preprocessing step, where approaches generally allow either discrete filtering of pixels classified as cloudy or accepting pixels with a low cloud probability as part of the signal [41–43]. As an explicit step, cloud masking has a long history combining feature engineering, thresholding and classical machine learning [44–46]. Recently, DNNs, especially CNNs, have increasingly enabled end-to-end cloud masking approaches without preprocessed features [47,48].

Completely delegating cloud detection to latent features of the same machine-learning model that is being used to learn the application features would be attractive due to simplified processing. Rußwurm and Körner [10] studied crop type mapping with Sentinel-2 top-of-the-atmosphere (Level-1C) time series data. They auspiciously observed that self-attention and recurrent based networks were able to suppress cloudy observations, although cloud masking still outperformed in the overall classification performance. Otherwise, explicitly assuming cloud detection as an latent hidden part of an end-to-end model has been relatively unexplored in previous work to the best of our knowledge.

Information about crop yields is needed on finer temporal and spatial resolution. The primary goal of this study is to find the most accurate, spatially scalable, yet operationally lightweight prediction method for statistical production. As explained above, this requires a solution that can effectively use long, irregular and sparse time series of EO data as basis of learning, and that is designed to provide predictions for fields of irregular size and shape. To this end, we (i) examine several approaches to construct spatial and temporal representations from the reflectance information of the observables. In addition, we explore cloud detection and crop yield in separate and combined scope by (ii) comparing the omission of cloudy pixels in preprocessing to end-to-end learning of cloud-contamination from data in our modeling schemes.

For reliable validation of crop yield mapping methods, accurate historical harvest data are needed. National agricultural statistics on crop yields are used as a primary source of reference data in many crop yield mapping studies e.g., [16,49–58]. However, in the global perspective, serious weaknesses have been identified in the practices of measuring agricultural production [59–61]. In Europe, regulation imposed by the European Commission streamlines statistical production for harmonized information at sub-national territorial levels. Crop statistics are usually based on farmer surveys. Nevertheless, in designing surveys, there is a trade-off between expenses and spatial coverage. Typically, nationally representative sampling frames place more weight on high production areas of the economically most important crops. As an implication, this leaves fringe regions with a paucity of information about crop yields and therefore, less accurate regional statistics. Acknowledging the effects of measurement error and sampling variability calls for more careful validation schemes for prediction models. We will additionally address this imperfection

by selecting highly representative municipalities in farmer survey data when assessing the accuracy of the method at the regional scale.

## 2. Materials and Methods

### 2.1. Study Area

Agricultural crop production in Finland is determined by the typical boundary conditions of high-latitude rainfed agroecosystems, namely, a short growing season, uneven rainfall distributions, special natural light conditions, and low growth temperatures, e.g., [62,63]. In focus in this study are all the main cereal crops, namely, spring-sown barley, oats, and wheat, and autumn-sown wheat, and rye [64]. The sowing period spans from the end of April in the south until early June in the north [65]. Winter crops are sown in August. Harvesting starts with winter crops in the early August followed by spring crops. Harvesting ends in September.

The study area as shown in Figure 1 comprises of 28 Sentinel-2 tiles that overlay approximately 92% of the arable land in Finland. The tiles were selected so that there were min. ~20 farms of any subsets (crop type per year) on a tile.

The growing conditions within the study area differ, because it extends from 60°N to 65°N. A recent study by [66] showed that there was substantial local and temporal variability in the average thermal growing season conditions in northern Europe. The main drivers of spatial variation were latitudinal and elevational gradients. The proximity of seas and lakes, and high forest cover, also typically characteristic of our study area in Finland, suppressed temporal trends and interannual variability [66].

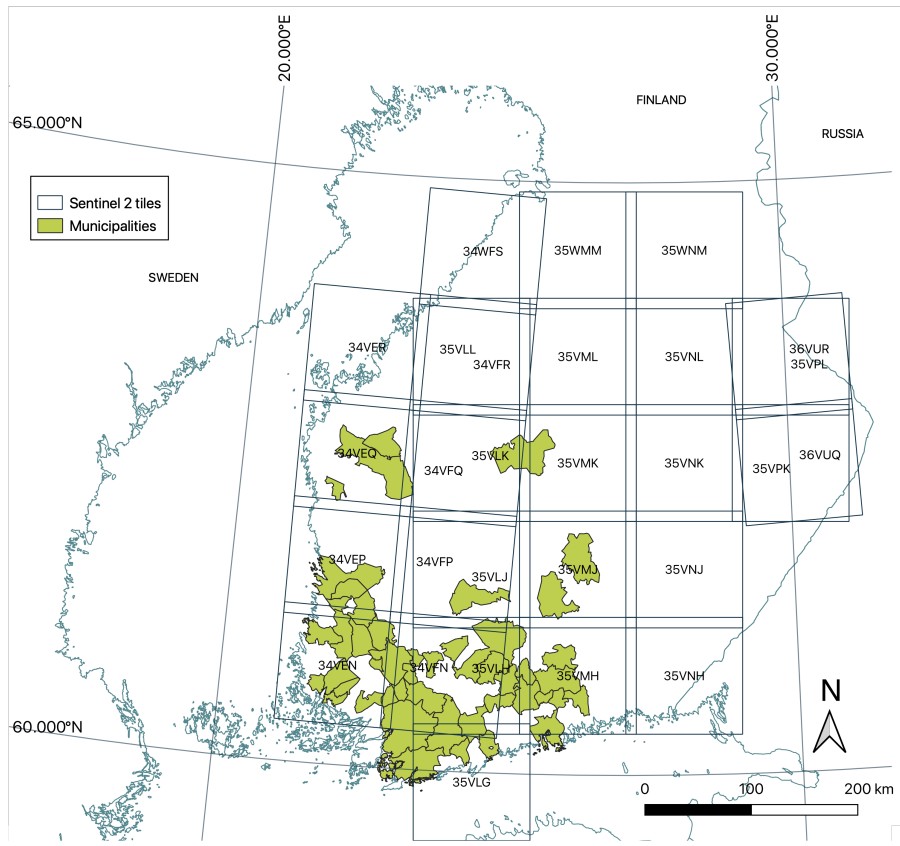

**Figure 1.** Study area is determined by 28 Sentinel-2 tiles. Selected regions (municipalities) are used in the study to validate regional-level yield predictions. Note that multiple Sentinel-2 tiles on adjacent orbits can overlay a municipality.

The variability in growing conditions is also shown in the crop statistics from the study area [64]: Figure 2 shows quite a high variation in regional crop yields, especially in the autumn-sown winter crops. Interannual variation is notably higher in the winter crops.

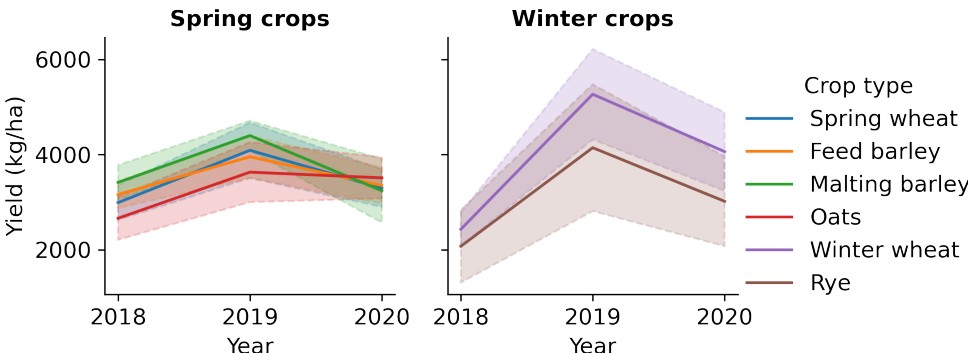

**Figure 2.** The mean and 68% bootstrap confidence interval (dashed lines) of the regional crop yields in the study area according to the crop production statistics [64] in 2018–2020.

## 2.2. Reference Data

The Finnish annual crop yield statistics are based on a farmer survey conducted by the Natural Resources Institute Finland (LUKE). We utilized the same survey data as a reference data. The sample unit is a farm, and thus, for each crop, we have the average yield at a farm level. The sampling design of the survey aims for an accurate estimate of crop yields at country level by following a multistage weighted sampling. Most of the weight is determined by the regional share of total harvested area for the main crops in Finland. We therefore do not have equal spatial coverage of farms in Finland, but most of the farms come from the high agricultural productivity regions in the southwestern part of the country. Other variables determining the weights in the sampling design are the production type and economic size of the farm.

Observed fields comprised approximately 6.5% (159,110 ha) of the total arable land in Finland. The main soil types in the sample were clay soils (54%), rough mineral soils (35%), and organic soils (6.5%). The proportions deviate slightly from the total sample; the Finnish cultivated soils are mostly 52%, 38%, and 10%, respectively [67]. The deviation can be explained by the sampling design, which favors the high agricultural productivity regions that are usually clayey.

Seeded crops are declared by farmers each year at the end of the sowing season (the middle of June) to the agricultural monitoring authority for being compliant for agricultural subsidies. Crop types and field geometries are stored in the Land Parcel Identification System (LPIS). We utilized LPIS data to select the subset of farms in the crop production sample growing the six main crop types (winter soft wheat, spring soft wheat, rye, feed barley, malting barley, and oats). The spatial observable unit was therefore a field (polygon) or a group of fields (multi-polygon) if a farm was growing the crop in question on several fields.

To ensure that all fields of a crop type are located in approximately similar agroclimatic conditions, we included only multi-polygons with an inner distance of less than 30 km. Additionally, we chose to filter out fields smaller than 1 ha in order to ensure an adequate number of pixels would represent a field. The average size of field parcel in Finland is rather small, namely 2.4 ha. In our remaining subsample, there were few farms growing a crop on a single parcel, therefore on an area of 1 ha only, but typically crops are cultivated on a larger scale on several parcels. In our subsample, the average area of a single crop per farm was 21.7 ha.

The number of farms in the crop-wise samples is shown in Table 1. With six crop types and three monitoring seasons, we have a total of 18 sample sets. Winter wheat and rye are winter crops and greatly differ from the summer crops in phenology. Winter

crops are sown in the autumn, and harvested earlier in August than summer crops. All varieties of crops have specific timing of developmental stages, in addition to the effect of agroclimatic conditions on their growth, and we therefore decided to develop separate models for each crop. However, feed barley and malting barley are phenologically similar, and in early-season crop forecasts the two varieties of barley are typically published in one yield estimate. We also have a combined barley model for in-season forecasting.

**Table 1.** Number of farms in the sample sets for all studied crops in 2018–2020.

| Crop Type | Year 2018 | Year 2019 | Year 2020 | Total |
|---|---|---|---|---|
| Winter wheat (*Triticum aestivum* L.) | 217 | 547 | 392 | 1156 |
| Spring wheat (*Triticum aestivum* L.) | 1895 | 1453 | 1544 | 4892 |
| Rye (*Secale cereale* L.) | 394 | 559 | 342 | 1295 |
| Feed barley (*Hordeum vulgare* L.) | 2633 | 2765 | 2629 | 8027 |
| Malting barley (*Hordeum vulgare* L.) | 707 | 396 | 486 | 1589 |
| Oats (*Avena sativa* L.) | 3095 | 3138 | 3445 | 9678 |
| Total | 8941 | 8858 | 8838 | 26,637 |

To validate regional-level yield predictions, we selected 91 subsets from 46 municipalities (see Figure 1) where the survey data comprises 30% of the whole cultivated area of a municipality (crop-wise). In addition, the minimum total cultivated area in the survey data subset was set to 200 ha. These municipalities were then considered to be adequately represented in the survey data, and the crop yields in these regions were calculated as the sample mean.

To assess the in-season prediction performance of our proposed methods, we used country-level crop forecasts from two sources as a reference. The European Commission's science and knowledge service Joint Research Centre (JRC) publishes European-wide model-based crop forecasts with the MARS Crop Yield Forecasting System (MCYFS). We collected forecasts for Finland from monthly MARS Bulletins (e.g., [68]). JRC publishes forecasts as early as in mid-June, and in mid-July and mid-August. LUKE is another source of country-level crop forecasts. LUKE's forecasts are based on regional agricultural advisors' estimates and they are published in mid-July and mid-August [69].

### 2.3. Optical Time Series Data

For crop yield prediction we needed time series of satellite observations of crop growth as an input for the prediction model. Geometrically and atmospherically corrected bottom of the atmosphere reflectance imagery (Level-2A) from the multi-spectral instrument aboard the Sentinel-2A and Sentinel-2B satellites were downloaded from the Copernicus Open Access Hub (Scihub). We excluded scenes with cloud cover over 95%. We utilized 10 spectral bands suitable for environmental monitoring with the following central wavelengths: Band 2 (492 nm), Band 3 (560 nm), Band 4 (665 nm), Band 5 (705 nm), Band 6 (740 nm), Band 7 (783 nm), Band 8 (842 nm), Band 8A (865 nm), Band 11 (1610 nm), and Band 12 (2190 nm) [70]. The observation period (10 May–31 August) covers most of the growing season in the study area. The average revisit-time over the study area is two days. Due to overlapping swaths from adjacent orbits, even more frequent observations are possible given cloud-free conditions.

Raw pixel values were extracted from the Level-2A product using a customized version of the EODIE toolkit (v0.1) [71] that builds on Python libraries. Pixels with the center point within the bounding polygon were included. The downstream modules of the processing pipeline included cloud masking, feature engineering, reshaping, and model training.

### 2.4. Cloud Masking

Using the scene classification map product from the Sen2Cor processor [72], we filtered out saturated or defective pixels, cloud shadows, clouds on medium and high probability,

and thin cirrus (classes 0, 1, 3, 8, 9, and 10). Cloud masking reduced approximately 55% of pixels (year 2020). Figure 3 shows the frequency distribution of reflectance values in 1000 bins for all bands from one season (10 May–31 August 2020) of field parcels growing cereal crops (oats, barley, wheat, rye) in the study area. The histograms in Figure 3a are calculated from unprocessed Level-2A data, and, in Figure 3b, after cloud masking. It seems that cloud masking has cut the long tail of high reflectance values. The bump-curves at the low reflectances of some bands are also somewhat dampened with cloud masking, but are not removed entirely. Otherwise, a visual inspection suggests the distribution curves very much resemble each other with or without cloud masking.

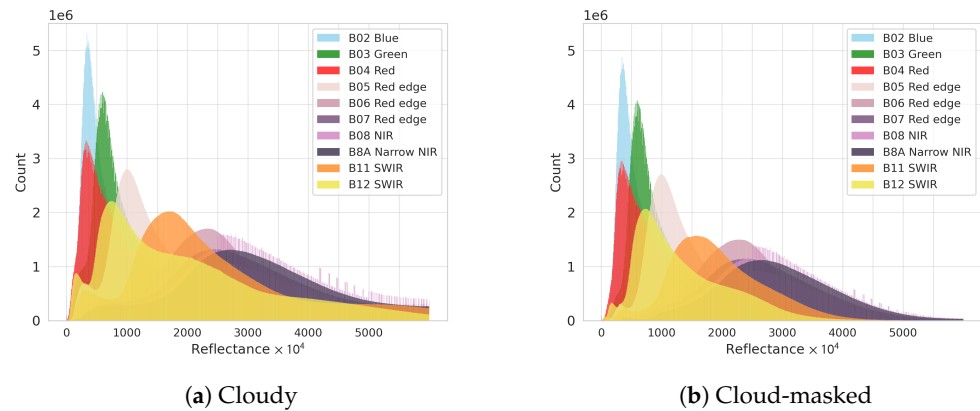

(**a**) Cloudy                    (**b**) Cloud-masked

**Figure 3.** Histograms of 10 Sentinel-2 bands with (**b**) and without (**a**) cloud masking. The data are from the growing season (May–August 2020) of all field parcels in the study sample. Note that the range of the surface reflectance values is set to [1, 6000]. The number of bins is 1000.

## 2.5. Object Representations

When monitoring the development of a specific crop type at farm-level, our sample consists of time series of observations from one or several field parcels. We had on average 2056 pixels representing a farm (1503 after cloud masking) at each time point. First, we treated each farm as an observational unit and constructed 32-bin histograms at each time point from 10 bands separately. In making the histograms, the range of values was determined by taking the 5th and 95th percentiles of the entire 2020 growing season cloud-masked reflectance distributions for each band separately. Figure 4 shows these upper and lower limits of range for bands 4, 8, and 12. The value range omits the small bump on the left and the long tails on the right. Figure A1 in the Appendix shows histograms with ranges for all 10 bands. The same upper and lower limits of range are used across all crop types.

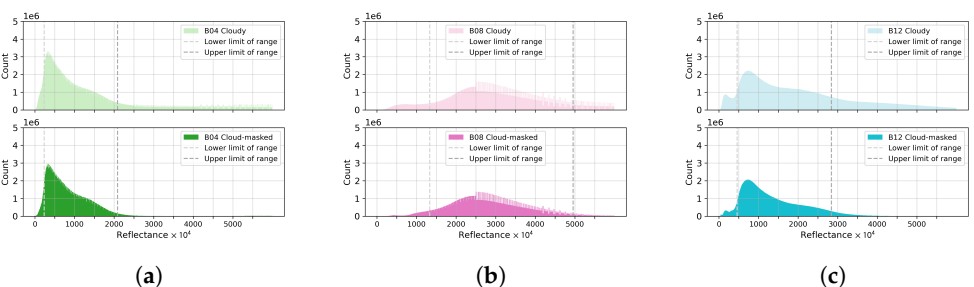

(**a**)                          (**b**)                          (**c**)

**Figure 4.** Displacement plots of surface reflectance histograms of three Sentinel-2 bands without cloud masking (top) and with cloud masking (bottom). The data is from the 11 May–31 August 2020. The range is set to [1, 6000]. The number of bins is 1000. (**a**) Band 4 (Red). (**b**) Band 8 (NIR). (**c**) Band 12 (SWIR).

The prediction models were also trained with median-based time series data for comparison. Figure 5 shows the median intensity values of spring wheat fields of a farm

during the 2020 growing season. The medians in Figure 5a are calculated from unprocessed Level-2A data and in Figure 5b after cloud masking. The bands seem to be correlated in both cases. Cloud masking seems to smoothen the time series curve. Before modeling, the median values were rescaled by dividing image digital numbers by 10,000, as shown in Appendix D in [73].

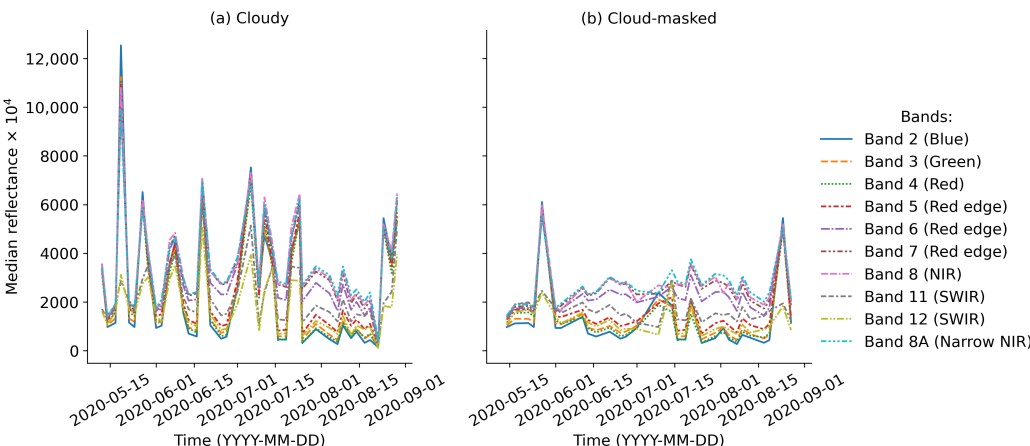

**Figure 5.** Median values of 10 Sentinel-2 bands from spring wheat fields of a farm calculated from (**a**) cloudy and (**b**) cloud-masked surface reflectance.

Figure 6 shows histograms of a farm during the 2020 growing season (spring wheat) from bands 4, 8, and 12. The data are $\ell_1$-normalized at observation dates. The dates without observations are simply zero-vectors. In some cases, cloud masking has resulted in completely excluding some dates due to clouds or cloud shadows. On average, there were 42 observation dates per season in the cloudy dataset, and 33 observation dates in the cloud-masked dataset.

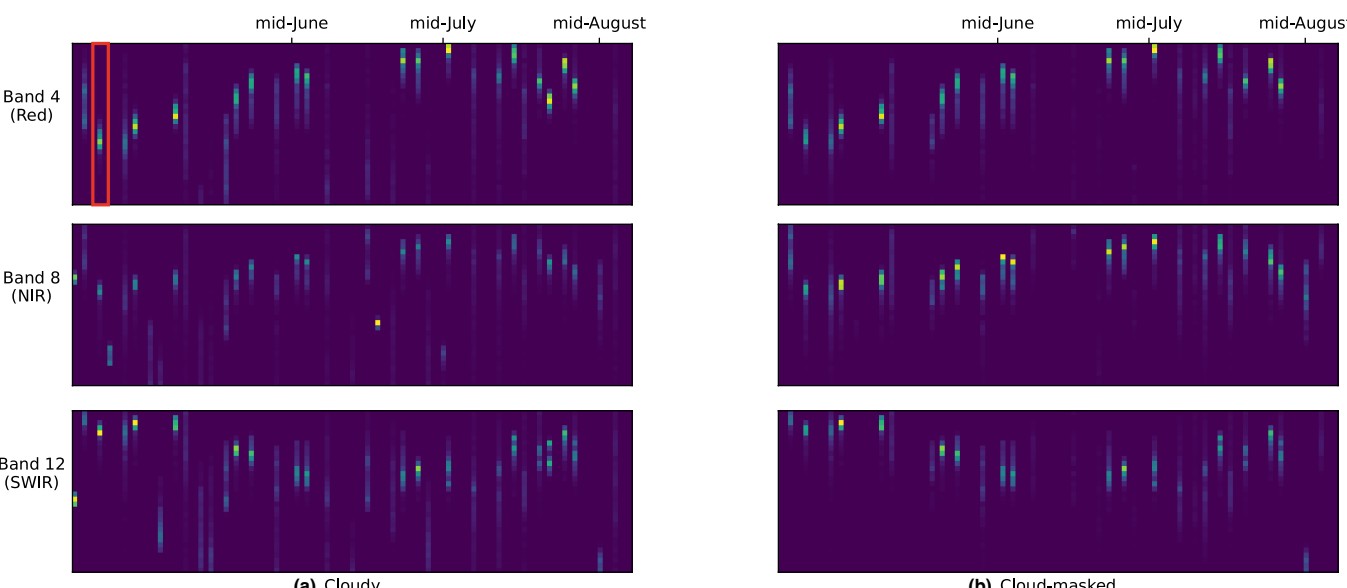

**Figure 6.** Histograms of three Sentinel-2 bands from spring wheat fields of a farm calculated from (**a**) cloudy and (**b**) cloud-masked surface reflectance. The x-axis is time, and the y-axis is the reflectance value range of the histograms (low values at the bottom, high values at the top). The red rectangle is an example of one timepoint in the early May, when the observed pixel values from spring wheat fields of a farm are shrunk into a 32-bin histogram. The brightest yellowish bins are the value ranges where the pixel counts are the highest.

We further explored the capabilities of histograms as density estimates of larger regions. From all the pixels of the survey data subset, we constructed municipality-level histograms. However, municipalities can expand over several Sentinel-2 tiles. In time series, there can therefore be observations from the adjacent orbits on different dates, posing a potential problem of over-represented observations from marginal areas. As a solution, we tested temporal compositing: Pixel values from an 11-day window were compiled into a histogram. The 11-day window was chosen because it is approximately the longest period of still modest phenological changes in crops, and at the same time, it is hoped that there exists at least one clear-sky observation day available for each sample with the period. Both farm-level (see Figure 7) and regional-level 11-day histograms were constructed. As the crop monitoring period was from the early May till the end of August, we had 12 windows per season. Similarly, we calculated the 11-day mean of median observations of farms (see Figure 8) and regions. In a few cases there were observations from only 11 windows due to clouds.

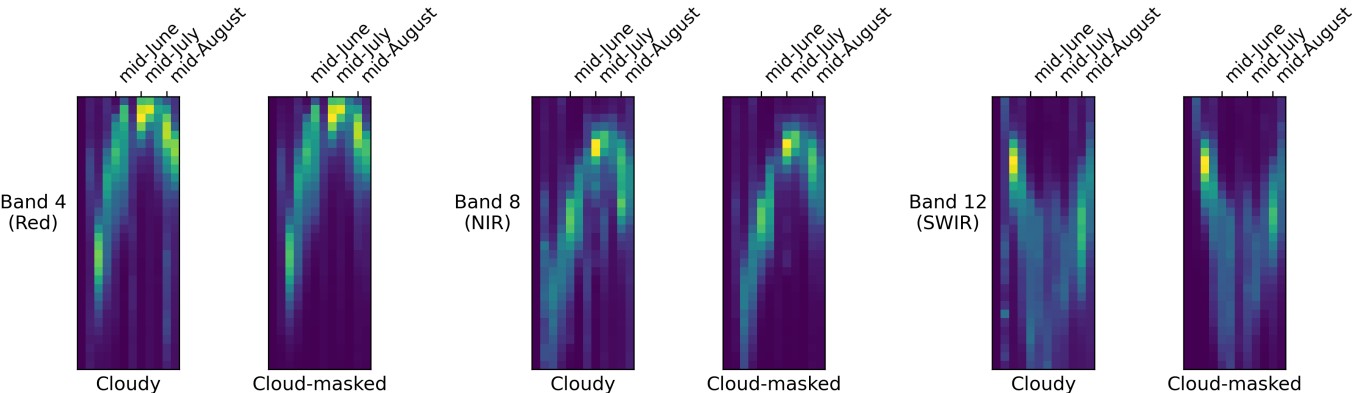

**Figure 7.** Histograms of three Sentinel-2 bands from spring wheat fields of a farm as 11-day composites. Calculated from cloudy and cloud-masked surface reflectance.

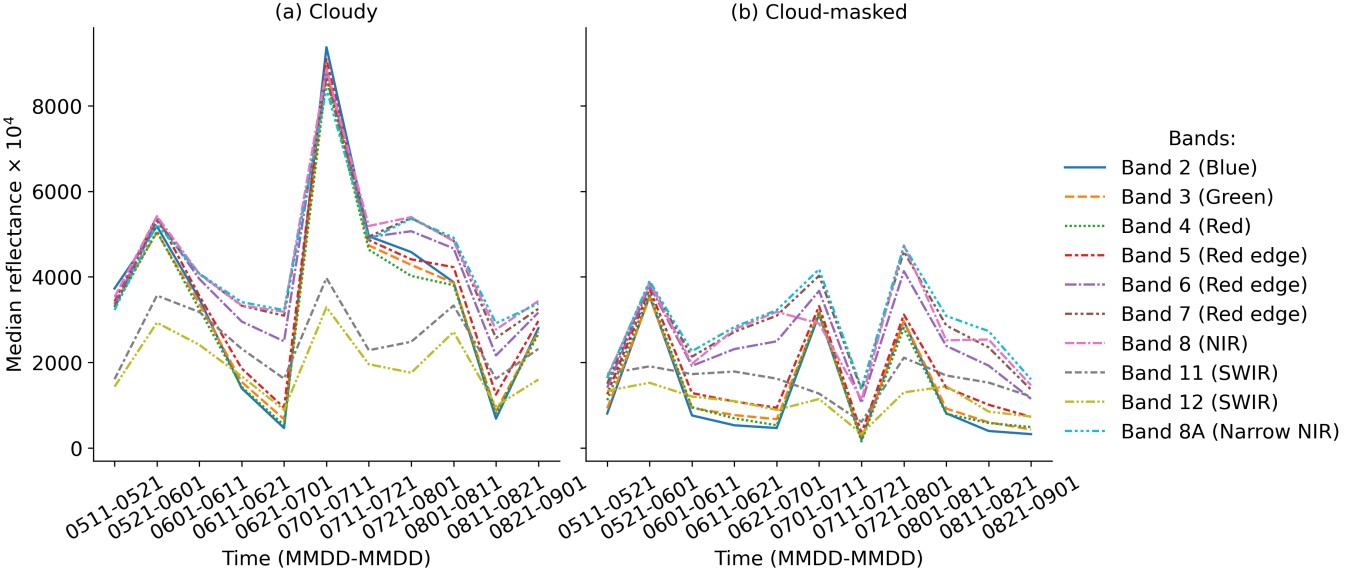

**Figure 8.** Median values of 10 Sentinel-2 bands from spring wheat fields of a farm as 11-day mean composites. Calculated from (**a**) cloudy and (**b**) cloud-masked surface reflectance. X-axis shows the 12 time windows from a growing season.

### 2.6. Prediction Models

For sequence modeling task we used RF as a state-of-the-art baseline method despite its limitations on time series tasks. We used RF implementation in sklearn-library version

0.23.2 [74] with parameters: 500 trees, and eight features randomly sampled as candidates at each split. The trees were grown to the maximal depth. For TCN, we used the Keras implementation, version 3.4.0 [75] in the Tensorflow environment (version 2.7.0) [76]. Keras TCN is based on [37]. TCNs were trained using the Adam optimizer [77] with parameters: learning rate $\alpha = 0.001$, $\beta_1 = 0.9$, $\beta_2 = 0.999$, and $\hat{\epsilon} = 0.1$. The batch size was set to 128. The validation loss was monitored with an early stopping mechanism. We used dilated causal convolutions for dilations 1, 2, 4, 8, and 16. The receptive field size was then 63. The validation split was set to 0.20.

### 2.7. Experimental Setup

We explored which machine learner and feature method performs best with farm-level data. For each crop, we trained a separate model, both with RF and TCN. We used two years for training and one year in testing. For cross-validation we iterated the training for 10 times. For each learner and feature method, we thus had performance metrics from 30 runs. For comparison purposes we chose to explore prediction accuracy in the mid-June, mid-July, mid-August, and at the end of the season (September 1). The TCN model can take a variable-size input, but RF was trained at each time point separately. From the model training perspective, we had either a $32 \times 10$ tensor (histogram) or a $1 \times 10$ tensor (median) for each time point to feed for TCN. For RF, the number of features was simply the time-steps $\times 32 \times 10$ (histogram) or time-steps $\times 1 \times 10$ (median). Missing observation days were padded with zero. We repeated the object representation schemes for both cloudy and cloud-masked datasets. The object representation schemes are illustrated in Figure 9.

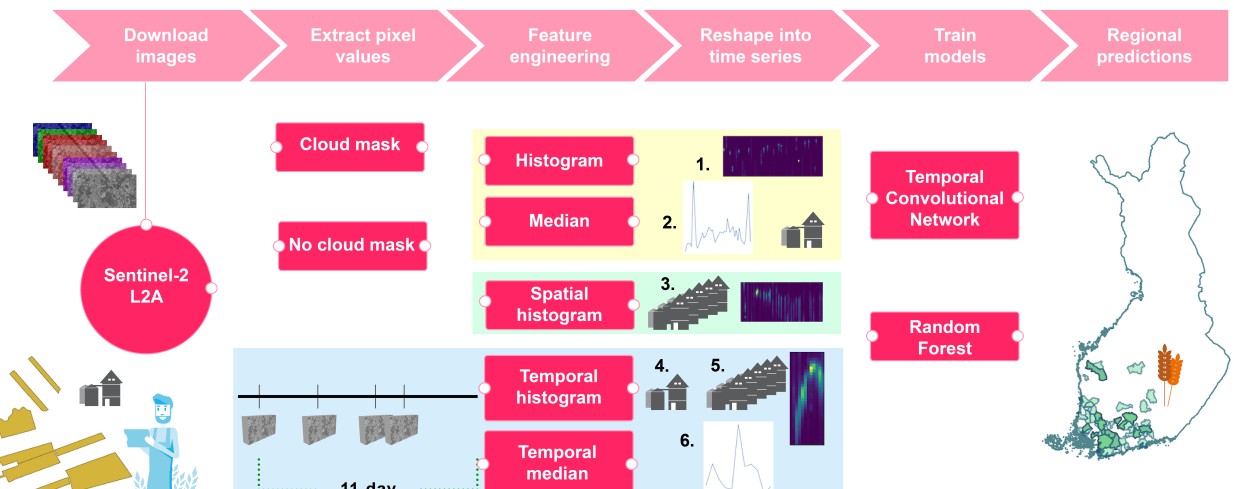

**Figure 9.** Processing pipeline to predict crop yield from satellite time series data. Six object representation schemes were applied as an input to temporal convolutional network and random forests. Object representation schemes were repeated with cloud-masked and not cloud-masked data. Models were trained at both farm-level and regional-level for comparison. Regional predictions were either mean-aggregated from farm-level predictions or taken from regional-level predictions.

## 3. Results

We start reporting the results with conventional cloud-masked datasets. See Table A1 in the Appendix for the performance metrics for all farm-level prediction models in the experiments. First, we compared the novel histogram-based TCN model with the conventional median-based RF as a baseline. Figure 10 shows the normalized mean of root-mean-squared error (NRMSE) and its standard deviation from the iteration runs with farm-level cloud-masked data. The accuracy of TCN improves in the course of the season, as is expected in time series prediction. RF predicts well in the early season, but the prediction

does not improve when adding new time features later the season. Overall, TCN has lower NRMSE than RF. Note that both models performed worse with winter wheat and rye (denoted with brown text). These are winter crops whose phenological development differs from spring crops.

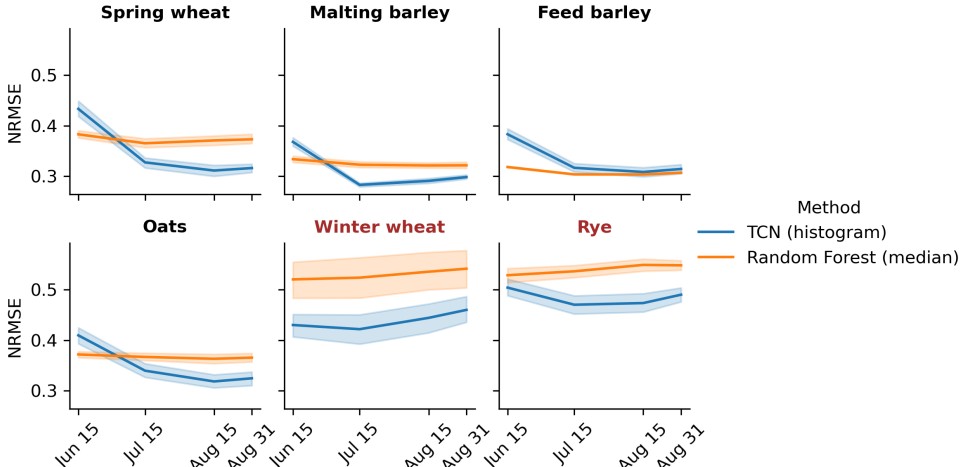

**Figure 10.** The normalized mean of root-mean-squared error (NRMSE), i.e., the share of root-mean-squared error from the mean crop-wise yield, and its 68% bootstrap confidence interval of farm-level predictions from four time steps within the growing season based on cloud-masked data.

As an example of how in-season TCN predictions evolve, Figure 11 shows yield predictions of a farm during the 2020 growing season (spring wheat) *vis-à-vis* the mean predictions and standard deviations of its surrounding farms in the same municipality. The blue vertical lines denote real observation dates, whereas missing dates are zero-padded. Both the individual farm prediction and the mean prediction of the region are quite close to the actual harvested yield from the mid-June until early July. In July, the predictions start to overestimate but again approach the true yield by the end of August. It is notable that the region prediction has clear in-season trends with moderate standard deviation. The trend is similar in both regional and farm prediction. This suggests that in these prediction curves, it may be possible to observe external factors in the growing conditions that determine the phenological development.

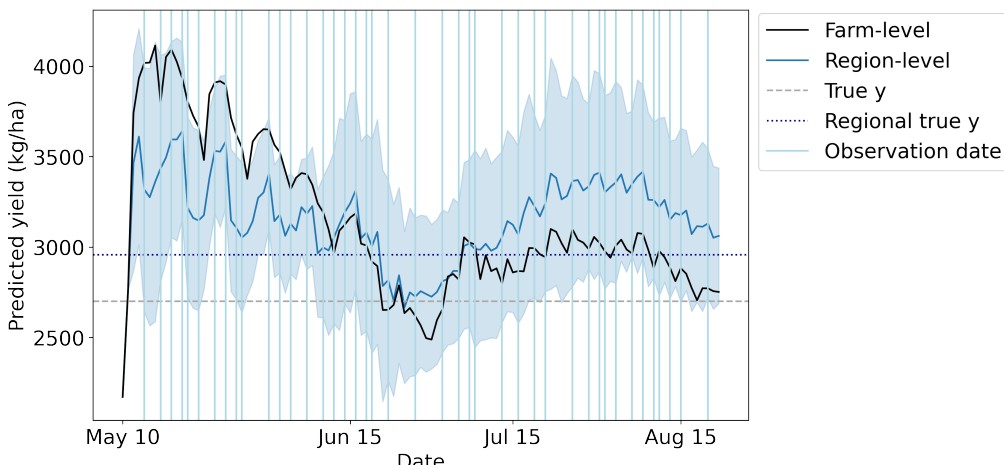

**Figure 11.** An example of in-season TCN predictions of spring wheat yield (kg/ha) of a farm and its surrounding region (municipality) in 2020 from cloud-masked data. Blue vertical lines denote observation dates. The region-level prediction is reported as the mean of all farm-predictions with the standard deviation.

Next, we calculated region-level histograms and their TCN predictions. Figure 12 shows the relationship of farm- and region-level predictions and actual spring wheat yields. In Figure 12a, the years are distinguishable. In the actual yields, there is great variation; in 2018 the yields were lower, in 2019 higher, and in 2020 average. The model overestimated the yields in 2018 and underestimated yields in 2019 and 2020. The area of spring wheat fields does not seem to make a visible difference. Note that there is vertical clustering of observations revealing rounding of reported yields. Figure 12b shows similar behavior to the model at the region level. The predictions tend to be conservative (closer to mean), and the area of an observation does not seem to be visually discriminative. The overall accuracy calculated in root-mean-square error (RMSE) is better at region level (729 kg/ha) than farm level (1131 kg/ha).

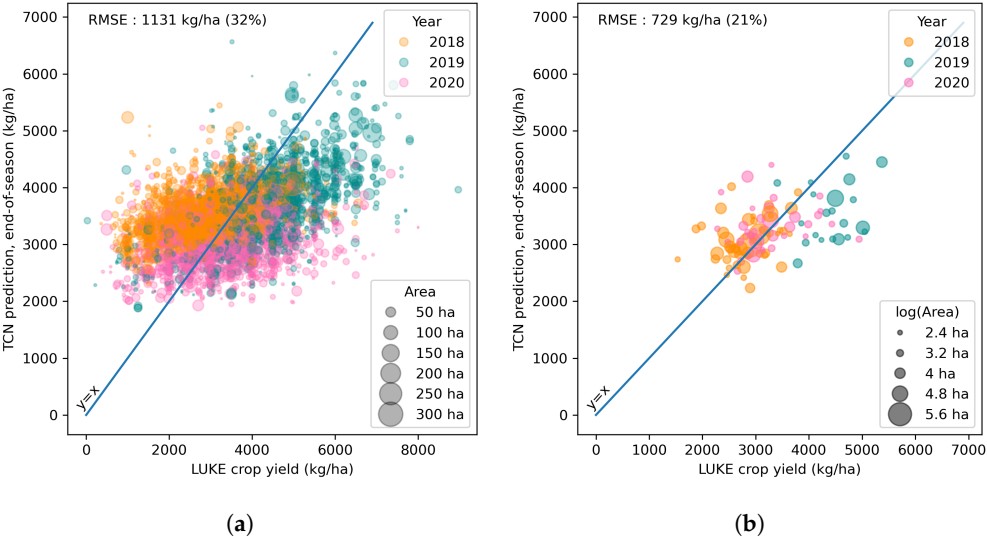

(**a**)           (**b**)

**Figure 12.** Scatter plots of farm- and region-level predictions vs. actual, when using histogram-based features with temporal convolutional network model, spring wheat, in 2018–2020, from cloud-masked data. The root-mean-squared error (RMSE) is shown in kg/ha and as a percentage from the average of the actual test set yield. The size of the point is proportional to the area of the observation (farm or region) in hectares. The years are denoted in colors. (**a**) Farm level. (**b**) Region level.

In Table 2, we have gathered several error estimation metrics of the iterated region-level crop yield predictions with 12 different feature engineering methods. Each dataset contains either an aggregated histogram of a municipality's pixels (region level) or mean-aggregated predictions from farms within the municipality (farm level). The datasets are either from non-cloud-masked or from cloud-masked data. The datasets are either from those municipalities that are overlaid by one or multiple Sentinel-2 tiles. The number of observations in the subset (region/crop/year) is either 495 if all regions are included, and 176 if only regions overlaid by one tile are included.

We use RMSE as the main metric for evaluating the quality of the solutions, since it provides the most direct interpretation in form of average error in kg/ha. We provide the average RMSE over all prediction tasks (different crop types and the three prediction scenarios corresponding to different years) as the aggregate summary, but additionally report the relative RMSE normalized with the mean regional true yields to even out the scale differences in yields across regions and crops. Mean average error (MAE) is provided as an additional aggregate summary. We also report the average correlation coefficient $\rho$ between the predictions and true yields, averaged over the different prediction tasks. For this metric we only consider cases with at least 10 samples, needed for reliable estimation of the correlation coefficient.

In terms of RMSE and MAE, the best prediction accuracy is achieved with farm-level histograms without cloud masking or temporal compositing. Similarly, high correlation $\rho$ agrees the result. Note that some model variants show even higher correlation indicating

that they can reliably rank the relative yields, but would be less useful for yield forecasting due to large average error. Indeed, temporal compositing seems to perform worse in all cases, indicating that we lose important information in the temporal compositing procedure. We expected to have superior results with regional histograms to farm histograms, especially in cases where the region falls under several tiles on adjacent orbits. However, the number of tiles does not seem to have importance here. Nor is cloud masking essential for good performance.

**Table 2.** The mean of root-mean-square errors (RMSE), the share of the error from the mean regional-level true yields (RMSE in %), mean absolute error (MAE), and the Pearson correlation coefficient $\rho$ of regional-level predictions with temporal convolutional network for 2018–2020. The best value per error metric is printed in boldface. The number (#) of tiles means that the dataset are either from those municipalities that are overlaid on one or multiple (1-6) Sentinel-2 tiles.

| Level | Feature Type | Cloud-Masked | # of Tiles | Temporal | RMSE (kg/ha) | RMSE (%) | MAE | $\rho$ |
|---|---|---|---|---|---|---|---|---|
| Farm | Histogram | | 1–6 | 121 days | **617** | **17** | **494** | 0.54 |
| Farm | Histogram | | 1 | 121 days | 642 | 18 | 540 | 0.67 |
| Farm | Histogram | x | 1–6 | 121 days | 709 | 20 | 585 | 0.56 |
| Farm | Median | x | 1–6 | 121 days | 728 | 20 | 570 | 0.50 |
| Farm | Median | | 1–6 | 121 days | 738 | 21 | 566 | 0.43 |
| Farm | Histogram | x | 1 | 121 days | 750 | 21 | 645 | 0.67 |
| Farm | Median | x | 1 | 121 days | 769 | 22 | 631 | 0.55 |
| Farm | Median | | 1 | 121 days | 809 | 23 | 631 | 0.54 |
| Region | Histogram | x | 1–6 | 121 days | 809 | 23 | 650 | 0.13 |
| Region | Histogram | x | 1 | 121 days | 841 | 24 | 692 | 0.10 |
| Region | Histogram | | 1–6 | 121 days | 862 | 24 | 676 | 0.11 |
| Region | Histogram | | 1 | 121 days | 909 | 26 | 739 | 0.06 |
| Farm | Histogram | | 1–6 | 11-day | 1006 | 28 | 809 | 0.62 |
| Farm | Median | | 1–6 | 11-day | 1006 | 28 | 744 | 0.45 |
| Region | Histogram | | 1–6 | 11-day | 1035 | 29 | 829 | 0.07 |
| Region | Histogram | x | 1–6 | 11-day | 1038 | 29 | 819 | 0.12 |
| Farm | Median | | 1 | 11-day | 1043 | 29 | 765 | 0.58 |
| Farm | Histogram | | 1 | 11-day | 1060 | 30 | 848 | 0.71 |
| Region | Histogram | | 1 | 11-day | 1094 | 31 | 885 | 0.00 |
| Region | Histogram | x | 1 | 11-day | 1126 | 32 | 904 | 0.07 |
| Farm | Histogram | x | 1–6 | 11-day | 1132 | 32 | 923 | 0.64 |
| Farm | Median | x | 1 | 11-day | 1161 | 33 | 903 | 0.58 |
| Farm | Histogram | x | 1 | 11-day | 1180 | 33 | 963 | **0.75** |
| Farm | Median | x | 1–6 | 11-day | 1193 | 34 | 938 | 0.50 |

Finally, we compared our best performing predictions with country-level crop yield forecasts published by JRC and LUKE (Figure 13). Here, we have used the best performing TCN model with a non-cloud-masked feature set. The actual yield levels have great variation between years. Overall, TCN predictions and forecasts alike tend to be more conservative and more likely to underestimate the yield compared to the actual yields. The TCN model is less accurate for winter crops (winter wheat and rye). To compare deviations from the actual yields, Table 3 shows that on average TCN outperforms published forecasts by 2.5 percentage points. If we include only forecast times in which JRC or LUKE forecasts are published, TCN outperforms the published forecasts by 2.2 percentage points.

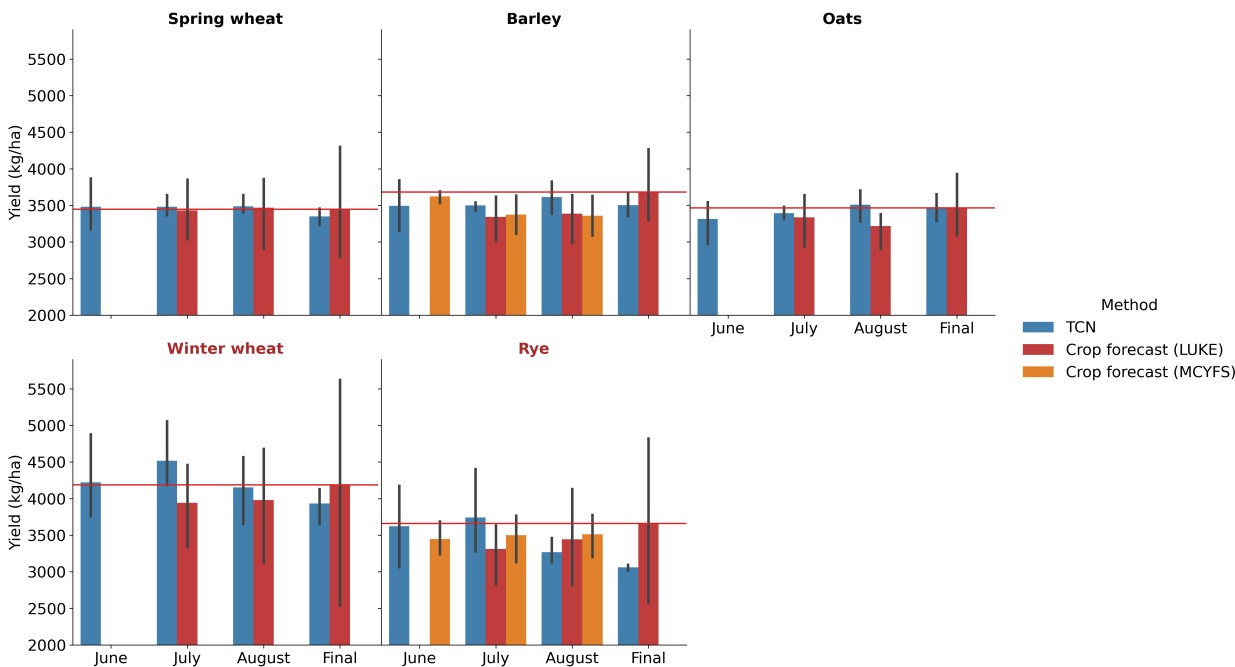

**Figure 13.** Bar plots of the mean farm-level predictions (TCN) and country-level published forecasts from JRC (MCYFS) and LUKE, in 2018–2020. The horizontal red line shows the 3-year mean of national level yields from the official crop statistics (LUKE). Black vertical lines show the minimum and maximum values of the three years. Note that the scale of the y-axis is zoomed to values between [2000, 5500].

**Table 3.** The deviation (%) of published country-level forecasts and TCN predictions from the actual crop yield. The lowest deviation (crop-wise) at the forecast time is printed in boldface. NaN means no forecast is available.

| Crop | Month | LUKE (%) | MCYFS (%) | TCN (%) |
|---|---|---|---|---|
| Spring wheat | June | NaN | NaN | 1.0 |
| | July | **−0.5** | NaN | 1.0 |
| | August | **0.7** | NaN | 1.2 |
| Barley | June | NaN | **−1.6** | −5.1 |
| | July | −9.2 | −8.3 | **−4.9** |
| | August | −8.0 | −8.7 | **−1.8** |
| Feed barley | June | NaN | NaN | −6.0 |
| | July | NaN | NaN | −3.9 |
| | August | NaN | NaN | −1.4 |
| Malting barley | June | NaN | NaN | 1.3 |
| | July | NaN | NaN | 3.0 |
| | August | NaN | NaN | 0.9 |
| Oats | June | NaN | NaN | −4.4 |
| | July | −3.7 | NaN | **−2.1** |
| | August | −7.2 | NaN | **1.2** |
| Winter wheat | June | NaN | NaN | 0.9 |
| | July | **−5.8** | NaN | 7.9 |
| | August | −4.9 | NaN | **−0.8** |
| Rye | June | NaN | −5.9 | **−1.0** |
| | July | −9.6 | −4.4 | **2.2** |
| | August | −5.9 | **−4.0** | −10.7 |
| Absolute mean deviation | | 5.5 | 5.5 | 3.0 |

## 4. Discussion

### 4.1. Prediction Performance

The growing conditions for crops at northern European high latitudes are in many ways exceptional in global perspective [62]. Weather conditions have large inter- and intra-annual as well as spatial variation [66] resulting in high natural variation in yields, which makes accurate prediction difficult. A similar study by Engen et al. [78] on farm-level crop yield prediction was located in Norway having somewhat similar cropping systems to our study area [66]. The amount of observations were also approximately comparable with ours. The best performing model was incorporating weather features and raw image data as 7-day composites into a CNN-RNN model. When farm-level yield predictions were scaled up to municipality-level, they reported nationwide RMSE of 308 kg/ha (compared to our 617 kg/ha). Unfortunately, NRMSE was not reported, making the results incomparable. However, interestingly, the study showed that raw satellite image data performed better than conventional handcrafted features (vegetation indices). Similar results were shown by [79]. Moreover, adding climatic data improved the performance.

Our results show that, the winter crop models perform worse on average than spring crop models, especially at the end of August. The winter crops winter wheat and rye have a different timing of phenological cycles than spring crops. Interannual and spatial variation in winter crops is also considerably higher than in spring crops as shown in Figure 2. As winter crops are harvested earlier, the model may have been confused about the post-harvest information in August. In addition, the amount of data in the winter crop subsets was lower than in other subsets, suggesting that with more training data available in the coming years, the model performance may improve. Another improvement to cover the high variability in growth conditions is to adjust the sampling design to ensure spatial heterogeneity of soil properties in the data.

For further improving the accuracy, one could consider fusion of other remote sensing data sources to cover the known growth factors. Grain yield in cereals is determined by the number of spikes per area, grain number per spike, and grain weight. These yield components evolve at different times of the growing season and therefore are exposed to different growth conditions and stresses, such as pathogens, pests, and plant nutrition, e.g., [80–82]. In crop yield studies, optical and near-infrared reflectance based vegetation indices have been widely used as a proxy for biomass accumulation e.g., [49,83,84]. However, by the senescence stage, vegetation indices are reported to carry less information for yield prediction, whereas climatic variables are more useful for monitoring yield affecting growing conditions and plant stresses [84,85]. In our study, setting we only utilize information from limited ranges of the electromagnetic spectrum provided by Sentinel-2. However, our deep learning based model readily supports the application of hyperspectral imaging in higher spectral resolution (higher feature space) to crop yield mapping when such data with appropriate temporal resolution becomes available. The prediction accuracy could also benefit from complementary information from microwave and thermal remote sensing, as suggested also by [86].

### 4.2. Reference Data

Lack of extensive farm-level or higher-resolution ground yields for empirical crop yield models is a common problem in large-scale yield mapping studies [8]. Even if higher-scale datasets exist in commercial agricultural systems, they are rarely leveraged due to privacy concerns. Similarly due to data protection and privacy laws, national statistical institutes can only publish aggregated yields. Typically statistics are published on administrative units. However, with satellite remote sensing, national statistical institutes could publish reliable temporally and spatially finer-scale yield forecasts. This would also provide other practitioners in crop forecasting a more precise proxy as a reference or validation for their operational systems.

The reference data notwithstanding poses a limitation to our study. There exists errors in the farmer-reported data, such as rounding (visible in Figure 12a). Furthermore, farms

that do not sell out the crops but use it as livestock feed, may only have a rough estimate of their crop yields. In addition, we note that the model is predicting harvested yield which is unavoidably smaller than the total yield, e.g., due to lodging or shattering. In our data we even had cases of harvest loss as high as 100%, i.e., farmers reported yields of value of zero. This implies crop was not harvested, e.g., due to overly wet harvest conditions.

In terms of potentially improved accuracy, our study would benefit from additional field-level training data, e.g., from yield monitors on-board combine harvesters, especially from data-sparse areas. If such an opportunity opens up our histogram-based approach readily allows fusion of data at different scales (one field or multiple fields of arbitrary sizes). Nevertheless, our results show that the farm-level data work sufficiently well in model training for regional yield forecasting. Therefore this work stands as a promising example for national statistical institutes typically holding historical farm-level survey data.

*4.3. Cloud Mask or Not?*

Cloud cover is a hampering factor in optical SITS tasks. It is challenging to detect various types of clouds: low and medium altitude water clouds; and high-altitude cirrus clouds in the upper troposphere and in the stratosphere. In addition, clouds cast shadows that result in darker reflectance areas. On the other hand, dark areas can be burned vegetation or topographic shadows. Clouds can also be similar in reflectance with a bright and white surface such as snow, ice, or water.

Several studies have evaluated the performance of cloud-masking algorithms on Sentinel-2 imagery. Typically, the studies evaluate the overall performance of the algorithms across several scenes. Indeed, cloud detection is challenging due to the high reflectance variability of earth surfaces. For example, [43] compared Sen2Cor, MAJA, LaSRC, Fmask, and Tmask in six very different scenes. They concluded that in overall, none of the algorithms outperformed the others. Sen2Cor exhibited the highest omission of clouds and shadows, performing better on clear-sky circumstances. In a similar comparison, for flat agricultural sites in Munich (Germany) and Orleans (France) [87] reported an inferior overall accuracy of Sen2Cor on two cloudy dates when compared with MAJA and FMask. Ref. [88] compared Sen2Cor, Fmask, and ATCOR on 20 Sentinel-2 scenes across all continents, different climates, seasons, and environments. They concluded that the overall accuracy was very high for all three masking codes and nearly the same for all algorithms. Based on their results, they suggested that Sen2Cor can best be applied for rural scenes in moderate climate and in scenes with snow and cloud.

From the application perspective, we aimed for routine use of Sentinel-2 data with a minimal computational burden and fluent automating of the whole deployment pipeline. We therefore chose to start with the Level 2A product, i.e., surface reflectance after atmospheric correction and its Sen2Cor cloud mask attached. We were encouraged by Sen2Cor's satisfactory performance in comparative studies. Ref. [10] had paved the way for further studies of learning cloud omission, and we tested our processing pipelines with and without cloud masking. Our results suggested that a deep learning based model can indeed learn to ignore cloud-affected data. A more-in-depth investigation is needed into how the model learns. Naturally, our results are restricted to relatively flat vegetated areas in a high-latitude boreal climate. Note that although the cirrus band 10 would be highly informative for the model to learn cloud-affected biases, it is only available in the Level-1C product (top-of-the-atmosphere reflectances). The Level-1C product could therefore be a more attractive target for further studying of end-to-end learning as showed by [10].

Another intriguing topic to study concerns how deep learned cloud omitting and separate cloud masks (such as Sen2Cor) is performing across all developmental phases during the growing season. The reflectance of the vegetation varies greatly through the growing season, especially when it reaches the senescence stage or at harvest time.

### 4.4. Object Representations

In the study, we constructed six different representations of the observations. At an overall level, we explored the capabilities of histograms as opposed to common median-based features. At farm level the results were ambivalent, meaning that median-based features occasionally outperformed histogram-based features. However, when focusing on regional-level results, histogram-based features performed better. A median-based approach yields 32 times fewer features for the model input. Classical machine learning algorithms would probably benefit from a smaller feature space. As with time, RF simply takes each time step as a new feature, and this also explodes the feature space. On the other hand, RF is famous for being invariant to irrelevant features, because trees are grown using random features [18]. It is therefore noteworthy that RF already shows its best prediction accuracies in the early summer, suggesting that early growing season features are truly most informative for the model.

Our results indicated that temporal compositing did not improve model performance, probably because of lost information. Yet it is possible that shorter window sizes in temporal composition would produce better results, especially if the area of interest is less cloudy. In any case, temporal compositing can be suitable for a visual characterization of the growing season and for easy fusion to other temporal data sources. In the present setting, the TCN can find more useful features in noisier data than in more preprocessed data.

### 5. Conclusions

The objective of the study was to find an operationally suitable method for large-scale crop yield estimations, so that we can produce in-season crop forecasts on an agroclimatically meaningful territorial scale. First, at farm level, we showed that the histogram-based TCN model outperformed the baseline median-based RF. Secondly, at region level, our results showed that TCN achieved the best prediction accuracy with farm-level histograms without cloud masking or temporal compositing. Our results indicated that cloud masking seemed to lose some information about the crop development. Similarly, temporal compositing did not improve model performance, probably due to lost information. Thirdly, at country-scale, TCN predicted in-season crop yield on average 2.5 percentage points more accurately than published forecasts from LUKE and JRC. In addition, we can produce forecasts for crops for which no in-season forecasts have earlier been published. We believe that more accurate regional level predictions as a reference can also boost the development of global scale crop monitoring.

**Author Contributions:** Conceptualization, M.Y.-H.; methodology, M.Y.-H.; software, M.Y.-H., S.W. and M.L.; validation, M.Y.-H.; formal analysis, A.K. and M.Y.-H.; investigation, M.Y.-H.; resources, M.Y.-H.; writing—original draft preparation, M.Y.-H. and M.L.; writing—review and editing, M.Y.-H., S.W., M.L., M.S., J.H., E.P. and A.K.; visualization, M.Y.-H.; supervision, P.P., J.H., M.S. and A.K.; project administration, M.Y.-H.; funding acquisition, M.Y.-H. All authors have read and agreed to the published version of the manuscript.

**Funding:** This research was funded by the European Union grant number 101037619.

**Informed Consent Statement:** Not applicable.

**Data Availability Statement:** The reference data used in this data originates from farmer survey and are confidential. The codes are available at https://github.com/myliheik/cropyieldArticle, (accessed on 1 July 2022).

**Acknowledgments:** Sentinel-2 imagery originates from the European Copernicus Sentinel Programme. We thank Anneli Partala (LUKE) for her continuous support and sharing of knowledge in crop production and crop statistics; Heikki Laurila (Statistics Finland) for invaluable commenting on the work from the outset; Mirva Kokkinen and Anna Auvinen (LUKE) for meticulous data curation; Jyrki Niemi (LUKE) for explaining global food market mechanisms; Ari Rajala (LUKE) for providing expertise in crop science; and Eeva Vaahtera and Jouni Hyvärinen (LUKE) for creating graphics worth a thousand words. We also acknowledge Kylli Ek and Johannes Nyman at CSC—IT Center for Science

Finland for technical support throughout the project. We made use of geocomputing tools provided by the Open Geospatial Information Infrastructure for Research (Geoportti, urn:nbn:fi:research-infras-2016072513) funded by the Academy of Finland, CSC—IT Center for Science Finland, and other Geoportti consortium members. Open access funding provided by University of Helsinki.

**Conflicts of Interest:** The authors declare no conflicts of interest. The funders had no role in the design of the study; in the collection, analyses, or interpretation of data; in the writing of the manuscript; or in the decision to publish the results.

## Abbreviations

The following abbreviations are used in this manuscript:

| | |
|---|---|
| EO | Earth Observation |
| CNN | Convolutional Neural Networks |
| DNN | Deep Neural Network model |
| JRC | Joint Research Center |
| LPIS | Land Parcel Identification System |
| LUKE | Natural Research Institute Finland |
| MAE | Mean Average Error |
| NRMSE | Normalized Root Mean Squared Error |
| RF | Random Forest |
| RMSE | Root Mean Squared Error |
| RNN | Recurrent Neural Networks |
| SITS | Satellite Image Time Series |
| TCN | Temporal Convolutional Network |

## Appendix A

*Appendix A.1*

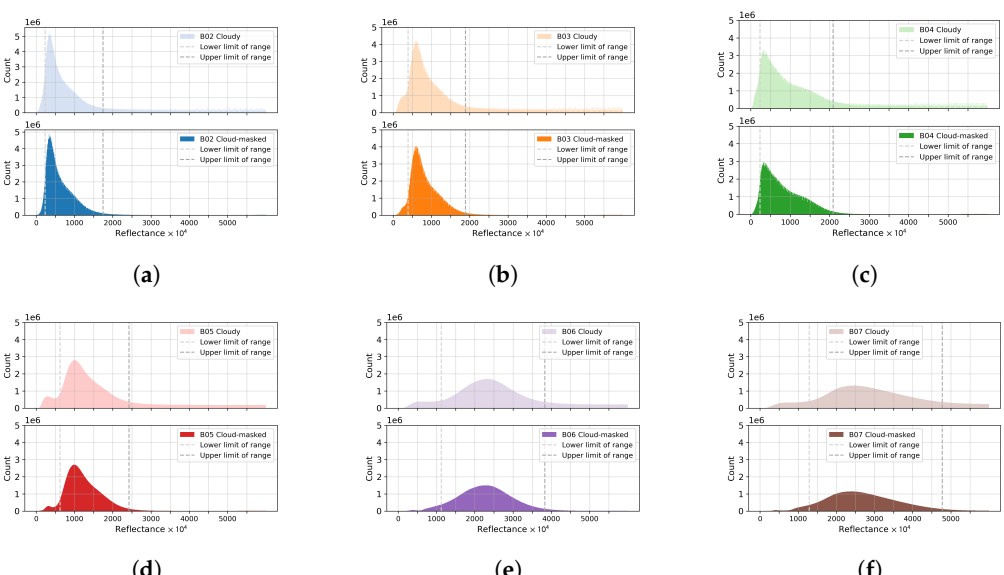

(a)  (b)  (c)

(d)  (e)  (f)

**Figure A1.** *Cont*.

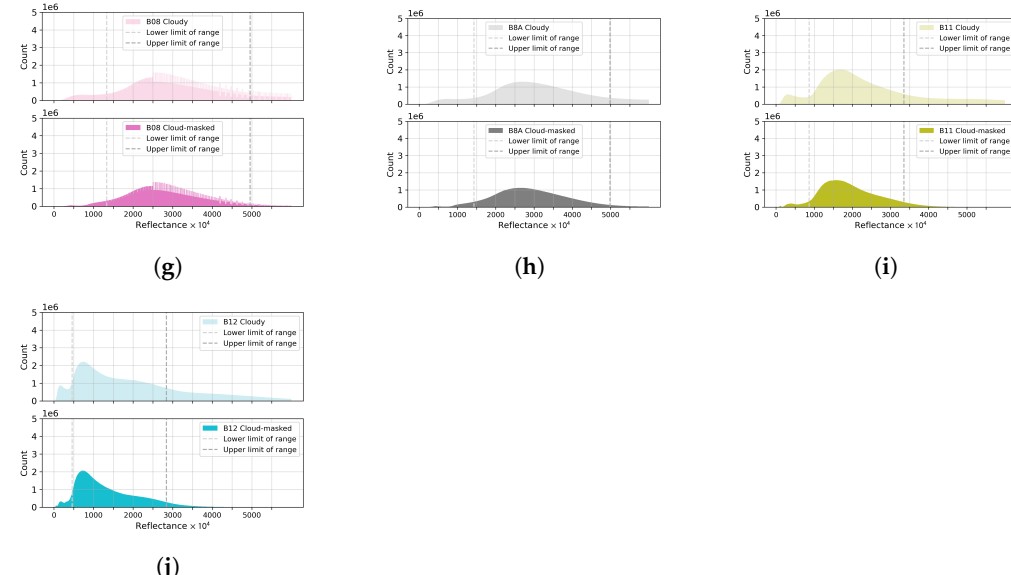

**(g)**  **(h)**  **(i)**

**(j)**

**Figure A1.** Displacement plots of histograms of all bands from raw BOA Sentinel-2 images before and after cloud masking. Range [1, 6000], bins 1000. (**a**) Band 2. (**b**) Band 3. (**c**) Band 4 (Red). (**d**) Band 5. (**e**) Band 6. (**f**) Band 7. (**g**) Band 8 (NIR). (**h**) Band 8A. (**i**) Band 11. (**j**) Band 12 (SWIR).

**Table A1.** The farm-level mean and standard deviations of root mean square errors (kg/ha) on test set validation with 10 training iterations of Temporal Convolutional Network (TCN) and random forests (RF) models when using crop type-wise subsets and median or histogram based features. The lowest mean error (crop-wise) at forecast time is printed in boldface.

| Crop Type | Method | Mask | June | July | August | End-Of-Season |
|---|---|---|---|---|---|---|
| Winter wheat | RF (histogram) | Cloud-masked | 1964 ± 383 | 1988 ± 386 | 2078 ± 432 | 2088 ± 427 |
| | | Cloudy | 1978 ± 384 | 1996 ± 393 | 2102 ± 443 | 2109 ± 442 |
| | RF (median) | Cloud-masked | 1912 ± 336 | 1914 ± 335 | 1979 ± 376 | 2003 ± 387 |
| | | Cloudy | 2001 ± 439 | 2027 ± 429 | 2090 ± 464 | 2072 ± 459 |
| | TCN (histogram) | Cloud-masked | 1629 ± 261 | 1544 ± 135 | 1637 ± 136 | 1735 ± 247 |
| | | Cloudy | 1822 ± 226 | 1594 ± 189 | 1673 ± 118 | 1754 ± 198 |
| | TCN (median) | Cloud-masked | **1476 ± 269** | 1489 ± 351 | 1576 ± 243 | 1630 ± 257 |
| | | Cloudy | 1553 ± 152 | 1513 ± 262 | 1659 ± 188 | 1749 ± 307 |
| | TCN (11-day histogram) | Cloud-masked | 2225 ± 801 | 1820 ± 573 | **1325 ± 180** | **1281 ± 180** |
| | | Cloudy | 2147 ± 464 | 2324 ± 985 | 1789 ± 602 | 1616 ± 436 |
| | TCN (11-day median) | Cloud-masked | 1539 ± 158 | **1396 ± 157** | 1405 ± 353 | 1459 ± 441 |
| | | Cloudy | 2326 ± 920 | 2459 ± 875 | 2429 ± 758 | 2571 ± 957 |
| Barley | RF (histogram) | Cloud-masked | 1127 ± 131 | 1105 ± 114 | 1112 ± 107 | 1115 ± 123 |
| | | Cloudy | **1125 ± 126** | 1109 ± 109 | 1115 ± 99 | 1119 ± 114 |
| | RF (median) | Cloud-masked | 1131 ± 147 | 1080 ± 112 | 1080 ± 104 | 1096 ± 137 |
| | | Cloudy | 1148 ± 136 | 1110 ± 118 | 1111 ± 112 | 1122± 139 |
| | TCN (histogram) | Cloud-masked | 1313 ± 136 | 1066 ± 93 | 1016 ± 73 | 1054 ± 88 |
| | | Cloudy | 1285 ± 162 | 1039 ± 103 | 1000 ± 81 | 1048 ± 117 |
| | TCN (median) | Cloud-masked | 1132 ± 109 | 980 ± 73 | **923 ± 62** | 998 ± 72 |
| | | Cloudy | 1174 ± 107 | **969 ± 112** | 936 ± 116 | **992 ± 146** |
| Feed barley | RF (histogram) | Cloud-masked | 1121 ± 115 | 1099 ± 96 | 1104 ± 87 | 1107 ± 102 |
| | | Cloudy | **1117 ± 109** | 1103 ± 91 | 1108 ± 83 | 1108 ± 95 |
| | RF (median) | Cloud-masked | 1128 ± 134 | 1074 ± 102 | 1073 ± 90 | 1087 ± 121 |
| | | Cloudy | 1146 ± 122 | 1109 ± 108 | 1107 ± 101 | 1114 ± 122 |
| | TCN (histogram) | Cloud-masked | 1343± 168 | 1108 ± 102 | 1077 ± 105 | 1100 ± 121 |
| | | Cloudy | 1309 ± 147 | 1030 ± 91 | 988 ± 60 | 1043 ± 92 |
| | TCN (median) | Cloud-masked | 1176 ± 115 | 1007 ± 91 | 963 ± 72 | **997 ± 85** |
| | | Cloudy | 1250 ± 114 | **993 ± 64** | **960 ± 58** | 1007 ± 64 |
| | TCN (11-day histogram) | Cloud-masked | 1340 ± 263 | 1541 ± 642 | 1258 ± 286 | 1188 ± 228 |
| | | Cloudy | 1205 ± 138 | 1206 ± 147 | 1088 ± 138 | 1070 ± 136 |
| | TCN (11-day median) | Cloud-masked | 1194 ± 104 | 1939 ± 1397 | 1490 ± 643 | 1348 ± 460 |
| | | Cloudy | 1466 ± 575 | 2021 ± 1057 | 1917 ± 858 | 1684 ± 630 |

**Table A1.** *Cont.*

| Crop Type | Method | Mask | June | July | August | End-Of-Season |
|---|---|---|---|---|---|---|
| Malting barley | RF (histogram) | Cloud-masked | 1253 ± 276 | 1217 ± 265 | 1229 ± 260 | 1234 ± 280 |
| | | Cloudy | 1268 ± 292 | 1214 ± 245 | 1219 ± 258 | 1220 ± 265 |
| | RF (median) | Cloud-masked | 1254 ± 289 | 1212 ± 272 | 1205 ± 262 | 1210 ± 288 |
| | | Cloudy | 1247 ± 294 | 1223 ± 292 | 1239 ± 322 | 1237 ± 330 |
| | TCN (histogram) | Cloud-masked | 1358 ± 182 | 1044 ± 101 | **1070 ± 80** | **1101 ± 110** |
| | | Cloudy | 1442 ± 186 | 1054 ± 112 | 1079 ± 80 | 1148 ± 146 |
| | TCN (median) | Cloud-masked | **1030 ± 76** | **1003 ± 101** | 1117 ± 195 | 1159 ± 232 |
| | | Cloudy | 1033 ± 65 | 1029 ± 101 | 1173 ± 246 | 1233 ± 272 |
| | TCN (11-day histogram) | Cloud-masked | 1399 ± 357 | 2267 ± 1111 | 1706 ± 634 | 1465 ± 440 |
| | | Cloudy | 1762 ± 650 | 1921 ± 1001 | 1834 ± 768 | 1717 ± 714 |
| | TCN (11-day median) | Cloud-masked | 1445 ± 346 | 1614 ± 489 | 1202 ± 223 | 1124 ± 163 |
| | | Cloudy | 1643 ± 761 | 1718 ± 993 | 1391 ± 551 | 1335 ± 468 |
| Oats | RF (histogram) | Cloud-masked | 1283 ± 49 | 1264 ± 27 | 1263 ± 24 | 1270 ± 30 |
| | | Cloudy | 1288 ± 42 | 1274 ± 30 | 1270 ± 30 | 1277 ± 35 |
| | RF (median) | Cloud-masked | 1264 ± 50 | 1244 ± 24 | 1230 ± 28 | 1239 ± 39 |
| | | Cloudy | 1294 ± 64 | 1283 ± 56 | 1273 ± 60 | 1273 ± 74 |
| | TCN (histogram) | Cloud-masked | 1381 ± 170 | 1140 ± 145 | 1069 ± 133 | 1090 ± 135 |
| | | Cloudy | 1450 ± 132 | 1147 ± 110 | 1071 ± 96 | 1092 ± 100 |
| | TCN (median) | Cloud-masked | **1203 ± 131** | **975 ± 81** | **904 ± 61** | **922 ± 71** |
| | | Cloudy | 1343 ± 175 | 1022 ± 142 | 978 ± 150 | 1025 ± 149 |
| | TCN (11-day histogram) | Cloud-masked | 1398 ± 170 | 1413 ± 389 | 1190 ± 205 | 1166 ± 165 |
| | | Cloudy | 1394 ± 190 | 1179 ± 151 | 1101 ± 87 | 1090 ± 99 |
| | TCN (11-day median) | Cloud-masked | 1487 ± 304 | 1702 ± 595 | 1422 ± 421 | 1370 ± 443 |
| | | Cloudy | 1726 ± 436 | 1855 ± 750 | 1482 ± 317 | 1453 ± 326 |
| Rye | RF (histogram) | Cloud-masked | 1850 ± 471 | 1884 ± 442 | 1923 ± 449 | 1919 ± 450 |
| | | Cloudy | 1853 ± 497 | 1876 ± 464 | 1919 ± 470 | 1911 ± 472 |
| | RF (median) | Cloud-masked | 1760 ± 386 | 1786 ± 370 | 1829 ± 375 | 1833 ± 390 |
| | | Cloudy | 1849 ± 423 | 1869 ± 422 | 1884 ± 433 | 1869 ± 462 |
| | TCN (histogram) | Cloud-masked | 1647 ± 199 | 1519 ± 123 | 1528 ± 116 | 1610 ± 223 |
| | | Cloudy | 1610 ± 222 | 1471 ± 234 | 1504 ± 165 | 1572 ± 218 |
| | TCN (median) | Cloud-masked | **1433 ± 291** | 1441 ± 289 | 1510 ± 291 | 1692 ± 332 |
| | | Cloudy | 1557 ± 323 | 1601 ± 336 | 1740 ± 333 | 1946 ± 357 |
| | TCN (11-day histogram) | Cloud-masked | 2103 ± 794 | **1431 ± 357** | **1332 ± 271** | **1344 ± 276** |
| | | Cloudy | 2586 ± 1586 | 2188 ± 1088 | 1761 ± 661 | 1702 ± 632 |
| | TCN (11-day median) | Cloud-masked | 1471 ± 287 | 1464 ± 361 | 1500 ± 345 | 1529 ± 412 |
| | | Cloudy | 1559 ± 407 | 1513 ± 363 | 1575 ± 504 | 1623 ± 527 |
| Spring wheat | RF (histogram) | Cloud-masked | 1310 ± 114 | 1284 ± 112 | 1316 ± 115 | 1317 ± 120 |
| | | Cloudy | 1316 ± 131 | 1290 ± 118 | 1324 ± 123 | 1329 ± 130 |
| | RF (median) | Cloud-masked | 1314 ± 121 | 1250 ± 114 | 1266 ± 106 | 1276 ± 118 |
| | | Cloudy | 1312 ± 145 | 1271 ± 122 | 1283 ± 124 | 1289 ± 138 |
| | TCN (histogram) | Cloud-masked | 1475 ± 193 | 1113 ± 96 | 1056 ± 83 | 1079 ± 92 |
| | | Cloudy | 1494 ± 228 | 1126 ± 140 | 1071 ± 113 | 1102 ± 156 |
| | TCN (median) | Cloud-masked | **1231 ± 171** | 1069 ± 119 | **1048 ± 157** | **1090 ± 196** |
| | | Cloudy | 1259 ± 174 | **1061 ± 157** | 1075 ± 156 | 1101 ± 189 |
| | TCN (11-day histogram) | Cloud-masked | 1843 ± 599 | 1960 ± 1021 | 1390 ± 334 | 1319 ± 261 |
| | | Cloudy | 1606 ± 252 | 1593 ± 575 | 1416 ± 305 | 1277 ± 213 |
| | TCN (11-day median) | Cloud-masked | 1990 ± 906 | 2155 ± 1254 | 1854 ± 702 | 1702 ± 588 |
| | | Cloudy | 1606 ± 348 | 1593 ± 586 | 1734 ± 482 | 1391 ± 231 |

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
