# Peer review of "Scalable Crop Yield Prediction with Sentinel-2 Time Series and Temporal Convolutional Network"

_remotesensing, doi:10.3390/rs14174193_

Round 1

Reviewer 1 Report

This paper aimed to investigate the possible use of a deep learning-based temporal convolutional network compared with the common machine learning algorithms such as random forest for predicting the crop yield using satellite image time series. The results of this study seem interesting and can be implemented by other scientists not only for predicting crop yield but also for other traits of interest. However, there are some points needed to be considered before the manuscript is accepted:

Keywords need to be modified as some of them are presented in the title, and it would be better to have other keywords than those in the title.

Line 8, “overall” would be better changed to “, in general,”.

I have never seen any introduction section split in the original research paper.

Too much information regarding Food security. Your study is more focused on using different algorithms to predict crop yield using satellite image time series to reduce food insecurity. However, you explained too much about food security but Ml/Deep methods/Remote sensing.

In the introduction section, you need to explain more about te current issue and problems, then provide a perspective on how to refine the problems. Also, you need to provide clearer examples of using Ml methods using remote sensing data to predict crop yield. There are good references for that, such as https://doi.org/10.3389/fpls.2021.777028 and https://doi.org/10.1038/s41580-021-00407-0.

The changes that you have seen in yield are not only coming from environmental factors but genetics. There is a possibility that new cultivars are used each year on those farms. The question is, how did you ensure the consistency of using cultivars in those farms over the years?

Did you collect the satellite images every day, every two days, every week? What was the time series interval for collecting images between May 10–August 31?

I could not find the evaluation parameters that you used to check the performance of the tested algorithms. Please provide at least two error estimation parameters such as RMSE, MAE, RRSE, etc. and two performance evaluation parameters such as R2, and r.

What was the number of CV you used in this study with 10 repetitions?

The discussion part needs to be improved significantly. You need to explain more about your research and compare the results with previous studies. Also, bring the strengths and the limitations of this research into the discussion and discuss them.

Please share the codes and datasets so that everyone can repeat the experiment.

Please check the paper for some grammatical and punctuational errors. 

Author Response

Point 1: Keywords need to be modified as some of them are presented in the title, and it would be better to have other keywords than those in the title.

Response 1: We now re-wrote the list of keywords to better reflect the content.

Point 2: Line 8, “overall” would be better changed to “, in general,”.

Response 2: We decided to keep the original word, which was suggested by the proofreading service to be in line with our choice of style and language (American English).

Point 3: I have never seen any introduction section split in the original research paper.

Response 3: We originally included the subsections in an attempt to clarify the story that covers several perspectives, but we agree that it is unusual. We now removed the subsections from the Introduction.

Point 4: Too much information regarding Food security. Your study is more focused on using different algorithms to predict crop yield using satellite image time series to reduce food insecurity. However, you explained too much about food security but Ml/Deep methods/Remote sensing.

Response 4: We start with food security to clarify the main motivation from the very beginning of the paper. Large-scale information on crop production is primarily needed for societal actors and international organisations, and focusing on this is what differentiates our work from yield prediction studies written in the context of e.g. precision agriculture where the main goal is maximal precision. In any case, we now made the food security section slightly more concise.

Point 5: In the introduction section, you need to explain more about te current issue and problems, then provide a perspective on how to refine the problems. Also, you need to provide clearer examples of using Ml methods using remote sensing data to predict crop yield. There are good references for that, such as https://doi.org/10.3389/fpls.2021.777028 and https://doi.org/10.1038/s41580-021-00407-0.

Response 5: We extended the references in Introduction (line 43 and line 130)  to provide a broader perspective to crop yield studies, and additionally expanded the Discussion (pages 13-15) to relate our work to previous work.
The technical motivation for our work is provided in the paragraphs of lines 55-61, 90-101 and 102-108, and can be summarized by three aspects: 1) Difficulty of addressing long, irregular and sparse time series; 2) how to handle fields and observational units of irregular size and shape, especially in a time series model, and 3) cloud occlusion. We now clarify the core technical elements in lines 120-122.

Point 6: The changes that you have seen in yield are not only coming from environmental factors but genetics. There is a possibility that new cultivars are used each year on those farms. The question is, how did you ensure the consistency of using cultivars in those farms over the years?

Response 6: We did not use (or have access to) information on cultivars or agricultural practices such as organic/conventional farming, since we need a model that can estimate the yield agnostic to possible external factors. Information on these aspects is not collected at the national level and hence practical predictive models will anyway need to operate without relying on the cultivars being identical across the years.

Point 7: Did you collect the satellite images every day, every two days, every week? What was the time series interval for collecting images between May 10–August 31?

Response 7: We collected all available Sentinel-2 images. The revisit time is about 2-3 days (line 231). Figure 11. shows an example of one farm from cloud-masked data. Blue vertical lines denote observation dates. On average there were about 30-40 observations per farm between May 10–August 31 as stated on line 274.

Point 8: I could not find the evaluation parameters that you used to check the performance of the tested algorithms. Please provide at least two error estimation parameters such as RMSE, MAE, RRSE, etc. and two performance evaluation parameters such as R2, and r.

Response 8: The exact configurations (hyperparameters) for the algorithms being tested are listed under Section 2.6: Prediction models (page 9).
We now complemented the results in Table 2 with mean absolute error (MAE) and the correlation coefficient r, and explain the choices in the main text. Since Table 2 presents the overall performance of the models across several different prediction tasks (predictions for different target years and different crops), the correlation coefficient is computed separately for each prediction task and the Table reports the average correlation across all tasks. We prefer correlation over R2 since the latter would be difficult to interpret in our case – it needs similar averaging but is influenced considerably more by the shift between the years, and hence would not quantify the differences between the methods as well.

Point 9: What was the number of CV you used in this study with 10 repetitions?

Response 9: We used two years for training and one year in testing. So 3 (years) x 10 repetitions (lines 303-305).

Point 10: The discussion part needs to be improved significantly. You need to explain more about your research and compare the results with previous studies. Also, bring the strengths and the limitations of this research into the discussion and discuss them.

Response 10: We modified the discussion accordingly. In Section 4.1. (lines 410-425) we explained our study setting and compared it with the traditional line of crop yield studies. We also compared our study to another similar study (Engen, 2020), explaining how we obtain prediction accuracies of similar magnitude but that the exact numerical results naturally cannot be directly compared due to different data and geographical area. To pinpoint the strength of our study, we now clarify that our deep learning based approach supports input data of higher feature space allowing the use of higher spectral resolution data or fusion of other data sources (line 422) into the model. We also clarify that the reference data poses a  severe limitation to the prediction accuracy as it includes inevitable errors (Section 4.2., lines 426-450).

Point 11: Please share the codes and datasets so that everyone can repeat the experiment.

Response 11: The reference data originates from farmer surveys and is confidential, as explained in the paper in the Data Availability Statement (lines 520-521). The farmer survey data is restricted to be used only for crop statistics and in-house developmental work related to crop statistics. Unfortunately, we cannot share these data sources. However, the code is already available at https://github.com/myliheik/cropyieldArticle

Point 12: Please check the paper for some grammatical and punctuational errors.

Response 12: We used a proofreading service before submitting our paper. We now reread the paper carefully and corrected the errors we found.

Reviewer 2 Report

This paper presents a method for estimating crop yield based on time-series satellite imagery, and compares the performance of the method under different features, different algorithms and different cloud processing operations to obtain an optimal yield estimation model. Overall the paper is a good paper in terms of fluency, logic and careful discussion of the experimental results, and I have only a few minor suggestions for the author.

Line 162-163: “Figure 2 shows quite a high variation in regional crop yields, especially in the autumn-sown winter crops. Interannual variation is notably higher in the winter crops.”

Figure 2 shows the mean and standard deviation of crop yields, generally the mean should be in the middle of the standard deviation, but this is not the case in Figure 2. Please recheck the Figure 2 or present the information in a table.

Line 311-312: “We repeated the object representation schemes for both cloudy and cloud-masked datasets. The object representation schemes are illustrated in Figure 9.”

In this study, the bands of Sentinel-2 images were used as input for model training. However, the information in the bands is limited and the authors may consider using some vegetation indices and image features as input to the model training, which may give better results.

Line 364-365: “TCN predictions and forecasts alike tend to be more conservative and more likely to underestimate the yield.”

The Figure 13 shows that TNC predicts a higher value than both LUKE and MCYFS, so the Figure should express that TNC is more likely to overestimate the yield of the crop. The authors are asked to reconfirm this section.

Author Response

Point 1: Figure 2 shows the mean and standard deviation of crop yields, generally the mean should be in the middle of the standard deviation, but this is not the case in Figure 2. Please recheck the Figure 2 or present the information in a table.

Response 1: We carefully re-checked the plots and can confirm that the mean is in the middle of the standard deviations, but that the error bands were indeed a bit difficult to distinguish from each other. We now slightly modified the visual style of Figure 2 to improve clarity by plotting the edge lines of the error bands in dashed-line. We feel that the visual illustration makes it easier to see the difference between the spring and winter crops (the latter having clearly higher variance), and since communicating this was the main goal of the illustration we decided to keep it as a plot; it is difficult to format a table that would convey the information clearly. 

Point 2: In this study, the bands of Sentinel-2 images were used as input for model training. However, the information in the bands is limited and the authors may consider using some vegetation indices and image features as input to the model training, which may give better results.

Response 2: We aimed at operational large-scale yield forecasting and hence preferred a model avoiding additional preprocessing steps, and as explained in Introduction (lines 118-123). Lines 83-90 argue that DNNs should be able to solve the problem without manual feature extraction. We agree that vegetation indices have shown good performance as model inputs and hence we now added a discussion element (lines 415-425) and references on this (lines 416, 419). 

Point 3: The Figure 13 shows that TNC predicts a higher value than both LUKE and MCYFS, so the Figure should express that TNC is more likely to overestimate the yield of the crop. The authors are asked to reconfirm this section.

Response 3: Even though our predictions are higher than those of the baselines, we actually still typically underestimate the yield. We now clarify this with “TCN and forecasts alike tend to be more conservative and more likely to underestimate the yield compared to the actual yields.” (line 362-363)

Round 2

Reviewer 1 Report

Agree